# Calling for Help—Peer-Based Psychosocial Support for Medical Staff by Telephone—A Best Practice Example from Germany

**DOI:** 10.3390/ijerph192315453

**Published:** 2022-11-22

**Authors:** Dominik Hinzmann, Andrea Forster, Marion Koll-Krüsmann, Andreas Schießl, Frederick Schneider, Tanja Sigl-Erkel, Andreas Igl, Susanne Katharina Heininger

**Affiliations:** 1Department of Anesthesiology and Intensive Care, University Hospital Klinikum rechts der Isar, Technical University of Munich (TUM), 81675 Munich, Germany; 2Association for Psychosocial Competence and Support in Acute Care-PSU-Akut, 81373 Munich, Germany

**Keywords:** health worker safety, mental health, peer support, psychosocial support, support by telephone

## Abstract

Background: A telephone support hotline (PSU-HELPLINE) was established at the beginning of the pandemic due to the burden on health professionals and the lack of support at the workplace. The aim of this study was to evaluate the telephone support service for health professionals in terms of its burden, benefits, and mechanisms of action. Methods: Data collection was conducted during and after calls by PSU-HELPLINE counsellors. In addition to the socio-demographic data evaluation, burdens of the callers and the benefits of the calls were collected. The content-analytical evaluation of the stresses as well as the effect factors were based on Mayring’s (2022). Results: Most of the callers were highly to very highly stressed. The usefulness of the conversation was rated as strong to very strong by both callers and counsellors. The PSU-HELPLINE was used primarily for processing serious events and in phases of overload. The support work was carried out through the following aspects of so-called effect factors, among others: psychoeducation, change of perspective, resource activation, problem actualization, connectedness, information, problem solving, self-efficacy, and preservation of resources. Conclusions: The expansion of local peer support structures and the possibility of a telephone helpline are recommended. Further research is needed.

## 1. Introduction

Coronavirus disease (SARS-CoV-2) has dramatically affected almost every country in the world. Evidence from previous epidemics shows that healthcare workers are at a high risk of developing both short-term and long-term sequelae. Specifically identified in the current and previous epidemics are anxiety, depression, fatigue and post-traumatic stress [1,2], sleep disturbances, relationship problems, behavioural changes (such as anger and substance abuse) [3], burnout [4], reduced psychosocial health [5] and psychiatric conditions [1,6].

The central component of any pandemic is the medical staff, both in acute care and in upstream or downstream medical care facilities, such as therapy facilities and nursing homes. However, not only since the times of the SARS-CoV-2 pandemic, but fundamentally occupationally, healthcare workers have been identified as particularly vulnerable occupational groups [7,8]. Serious events that occur in the context of acute care can result in trauma among medical staff. Thus, medical staff can become victims [9]. The term “second victim” was introduced by Albert W. Wu [10] in 2000. The term “second victim”, which is now often also referred to as “second affected”, was originally used to describe staff who were traumatised by self-inflicted treatment errors. Scott and colleagues expanded the term to include healthcare workers who may also be traumatised by unforeseen patient care incidents and become ill as a result [11].

The well-being of healthcare workers is not only important for them and their families, but also has a significant effect on the quality of patient care and patient safety. On a systemic level, the result of a systematic review and meta-analysis is: “Physician burnout is associated with poor function and sustainability of healthcare organisations” [12]. Thus, there is a crucial key point in the entire healthcare system [4,13]. According to Drudi and colleagues 2021 [14], “well-being” and job-related burnout symptoms are increasingly seen as important forces in the retention or loss of healthcare workers. Although the above-mentioned consequences are well known, there is little systematised data on the burden of disease with regard to psychosocial illnesses and dysfunctions among medical staff [15].

### 1.1. Psychosocial Support

In this context, structures are needed to protect the risk group “medical staff” as best as possible from their own illnesses because of patient care. The structures of standard medical and psychotherapeutic care are often only noticed by the risk group at a very late stage or not at all. The reasons for this can be the following: presumed “loss of face” when helpers themselves need help, worries about stigmatisation, difficult acceptance of one’s own illness or also shame and worries about not being up to the job [16,17]. One possibility of low-threshold support, especially for medical staff, and as an interface and ice breaker into standard care is so-called peer support [15,18]. For this purpose, suitable colleagues are trained in psychosocial support so that they can then work as so-called peers [9]. On the one hand, this can take place within one’s own team, for example, within the clinic, but also through higher-level structures, independent of the institution. Both forms have their advantages. Team-internal peers notice at an early stage when colleagues show abnormalities (e.g., irritability, lack of concentration, limited ability to work, sleep disorders, etc.) due to serious events in their daily work and show a need for psychosocial support. The advantages of higher-level structures are that support can be provided without the knowledge of direct colleagues and superiors. This means that a high level of anonymity can be offered, for example, when support is offered by telephone. This possibility of anonymity can be of great importance to those affected for various reasons: for example, because (1) the internal team structures are not considered suitable for one’s own stress; (2) the person does not yet show any conspicuousness within the work, but has nevertheless identified a need for support for him/herself; or (3) the person holds a leadership position, and because of this, does not want to reveal any perceived weakness to colleagues.

Collegial support in stressful situations (e.g., after medical errors) has been used successfully for some time, especially internationally, and independently of the pandemic [10,19]. In part, positive experiences from other areas, such as the US military in dealing with Vietnam returnees with post-traumatic stress reactions, were transferred from Critical Incident Stress Management (CISM) according to Jeffrey T. Mitchell (“Battle Buddies”) [20]. Among others, the work of Drudi and colleagues (2021) [14] emphasises the importance of peer support structures. In this paper, the focus was on vascular surgeons and residents in the specialty, in contrast to many other papers that dealt predominantly with nurses. David and colleagues (2022) [21] reported that a strong community and camaraderie structures to achieve common goals are particularly efficient and effective in helping individuals cope with shared traumatic experiences. Peer support systems for health professionals can be a successful method to minimise or overcome existing access problems to psychosocial support [9,22].

So far, there are no nationwide structures of psychosocial support by peers in Germany. A survey among anaesthesiologists in Germany shortly before the start of the SARS-CoV-2 pandemic showed that both a basic need for psychosocial support was explicitly expressed and low-threshold support by colleagues was explicitly desired [23].

### 1.2. Anonymous Psychosocial Support by Telephone—Study Situation

In view of the needs mentioned and the possibilities for support identified, the following section looks specifically at psychosocial support by telephone for medical staff in Germany.

The currently available studies on telephone psychosocial support for healthcare personnel, and thus, on its effectiveness, appear to be limited. A systematic literature analysis according to the PICO model was only able to identify a Cochrane Review by Pollock and colleagues (2020) [24] within the search criteria, which includes the presentation of four relevant studies on telephone support measures. These are briefly presented below.

In France, a psychological support programme exists for 39 hospitals based on a hotline system (certified psychologists) with a medical backup (psychiatrist). Feasibility (149 calls in 26 days) as well as dissemination (different specialties and departments in hospitals) and usefulness have been demonstrated (70% of callers in terms of SARS-CoV-2 and other support) [25].

Maunder and colleagues (2003) [26] (Mount Sinai Hospital, Toronto, ON, Canada): Confidential telephone hotline (psychiatric nurses) for all hospital staff. The programme was found to be particularly effective for those in quarantine. In addition, an informal network of telephone contact and support was offered by quarantined ICU nurses (“telephone-supportline”, “informal network of mutual telephone contact and support”). An important insight from this is that the crucial point is probably not to feel alone.

Feinstein and colleagues (2020) [27] (Austin, TX, USA): “The Healthcare worker mental health COVID-19-hotline”. This incorporated the expertise of psychiatrists, clinical social workers, and mental health volunteers. It was a 13-step plan to develop a hotline. The main objective of the work was to present and promote the implementation of the telephone service and the dissemination of the developed approach. This was mainly to maintain and optimise the manpower and capacity of medical staff.

Gonzalez and colleagues (2020) [28]: Spiritual support hotline on the part of the hospital chaplain and centralised support helpline (The Department of Psychiatry and Behavioral Health). This can be accessed by medical staff and their family members to provide direct consultations and other options (9 a.m. to 5 p.m.). Voicemails left by callers are answered within 24 h. Most important message: “We’re in this together!”.

In summary, positive effects of telephone psychosocial support can be shown in all four examples. Valid conclusions from RCTs are not yet available.

In the following, a best practice example from Germany is presented: PSU-HELPLINE (PSU = German abbreviation for psychosocial support). The focus is on telephone-based psychosocial support for medical staff by peers.

### 1.3. PSU-HELPLINE—Psychosocial Support by Telephone

Since low-threshold offers in the form of peer support systems were rare in Germany at the time of the start of the pandemic [15] and hardly available in the form of telephone help, the non-profit association PSU-Akut introduced a telephone peer support service, the so-called PSU-HELPLINE, for quick help. Since then, this service has been offered as a free-of-charge, confidential, and anonymous telephone counselling service for employees and managers from the health and emergency services. The aim is to preserve the health and ability to work as well as to explicitly stabilise and restore the ability to act in highly stressful situations, of medical staff. The PSU-HELPLINE offers the possibility to transfer people with an explicit need for treatment to integrated, initial structures of standard psychotherapeutic care in a low-threshold and timely manner. The service offered is financed by various healthcare providers in Bavaria, Germany, and by a foundation. The psychotherapists, the psychotherapeutic head as well as the coordination of the PSU-HELPLINE are employees or freelancers with approbation. The counsellors of the PSU-HELPLINE work on a voluntary basis and receive an expense allowance.

Peers provide counselling on the PSU-HELPLINE telephone. The team of counsellors consists of doctors, nurses, medical assistants, and rescue service staff who have completed additional training in psychosocial support (peer support) or psychotraumatology. To be a counsellor at the PSU-HELPLINE, many years of experience in psychosocial emergency care is a prerequisite. This can be acquired in Germany through various training institutions. Furthermore, only consultants who are themselves active in the healthcare sector are selected.

The counselling includes, if needed, first a telephone intervention as well as counselling based on psychosocial support measures including the provision of professional information.

Through a peer as a discussion partner, health workers can talk to a counterpart who is familiar with similar situations and stresses and does not need any supplementary professional explanations, as would be necessary in a discussion with counsellors from outside the field. This aspect is the central element of the peer approach: the same among equals. The counselling sessions provide a framework for callers to clarify current stresses, provide relief, receive information on stress and stress reduction, and develop strategies for stabilisation and coping. Together with the peer, it can be decided whether and what further specialized support is necessary and which counselling centers can offer additional help. A maximum of three conversations can be held with a peer.

If necessary, for example, in the case of psychological trauma, the peer can be referred to a psychotherapeutic telephone session on the same day of the call. Here, psychotherapeutic crisis intervention measures and up to five conversations are offered. In addition, there is the possibility of directing those affected further into standard psychotherapeutic care via the German so-called psychotherapist procedure of the accident insurance funds. In case of emergency, the PSU-HELPLINE counsellors have a direct line to the psychiatric crisis service.

In addition to individuals, managers and personnel managers can also use the PSU-HELPLINE to obtain telephone counselling for systemic psychosocial support for work units (e.g., teams/wards) and institutions. Managers thus not only can contact the PSU-HELPLNE in case of their own stress but can also get information on help and support for their stressed teams. If necessary, further measures in the form of on-site interventions for medical teams can emerge from discussions on systemic support options.

Based on the previous considerations, results from the evaluation of the PSU-HELPLINE are presented below. The following questions are the guiding principles for this.
(1)How burdened do the callers feel and how helpful do they find the conversation on the PSU-HELPLINE?(2)Which burdens do the callers mention?(3)How can a telephone support service reduce the psychosocial stress of the callers?

## 2. Materials and Methods

### 2.1. Design, Setting and Data Collection

The present study was conducted in a cross-sectional design. Data collection took place between April 2020 and April 2022.

Two survey instruments were used: a meta-questionnaire and a short questionnaire. Both questionnaires were completed by the PSU-HELPLINE counsellors during and after the call. The callers could give their consent to data collection and data processing by telephone. Only data records for which consent to data collection was given were included in the results. The number of calls thus differs from the number of data records that can be analysed.

The meta-questionnaire was filled out directly after the interview. It was mainly used to document the call and for quality assurance. The short questionnaire served, on the one hand, to support the counsellors during the conversation; on the other hand, the following aspects were recorded: items on socio-demographic data, current stress (assessment by the counsellor and about the caller, 5-point Likert scale), relief from the counselling interview (5-point Likert scale). After the conversation, the counsellors noted down the thematic content of the conversation in a free-text format. The conversations were not recorded. They are memory transcripts and not verbatim transcripts.

The answers were evaluated descriptively as well as content-analytically.

### 2.2. Study Participants

The sample consisted of callers to the PSU-HELPLINE during the specified survey period. Each call to the PSU-HELPLINE telephone number was recorded anonymously as part of their consultation. The PSU-HELPLINE counsellors asked the caller whether the data could be recorded anonymously for documentation and evaluation. All people who agreed to this procedure were included in the sample.

### 2.3. Research Ethics

Calls to the PSU-HELPLINE are anonymous and confidential. Data collection is guided by the research-related ethical guidelines of the German Psychological Society [29].

To protect their data and to consider the aspects of voluntariness and informedness, the callers were actively asked by the counsellors for permission to store the data and informed that they would not suffer any disadvantages if they did not agree to the data collection. In accordance with the regulations of the German Data Protection Ordinance, the option to enter a pseudonymisation code was given. There were no risks involved in answering the questions and the studies did not contain any manipulations. The collected data were stored on password-protected drives.

Over the duration of the project, the surveys were conducted during the process. The support and care of the target group were always explicitly in the foreground. Both evaluations and research projects were subordinated to this aspect at all times in order to protect the needs and trust.

An ethics vote (683/20 S-SR) of the ethics committee of the Klinikum rechts der Isar of the Technical University of Munich is available for accompanying the research of PSU-Akut e.V. measures.

### 2.4. Bias

Common-method bias with regard to the stress scale could be countered by the simultaneous use of self- and other-assessment scales. Since the questionnaire is not a classic data collection instrument but was filled out by the counsellors instead of the respondents, further method biases were not addressed. The limitations resulting from this are explained in the discussion.

### 2.5. Study Size

As the study was conducted as an evaluation of a collegial psychosocial support service and no statistical tests requiring a priori test strength analysis were planned, this was not conducted.

### 2.6. Statistical Methods—Quantitative Part

All analyses were conducted using IBM SPSS Statistics 26 (Ehningen, Germany). After data cleaning, descriptive statistics analyses were performed.

### 2.7. Content Analysis—Qualitative Part

The data basis for the qualitative analysis of the data was an open question within the short questionnaire with the focus on “interview content”. The evaluation of the data was carried out by means of qualitative summary content analysis following Mayring (2022) [30].

## 3. Results

### 3.1. Structure of the Sample

A total of *n* = 348 data sets were included in the analyses. The following data include all responses without missing values.

The callers to the PSU-HELPLINE were 83.2% female and 52.5% aged between 30 and 50 years. Doctors were the largest professional group of callers with 30%. A total of 76.9% of the callers came from southern Germany.

The detailed structure of the sample can be seen in Table 1.

### 3.2. Descriptive Data—Calls

Calls lasted an average of 33.56 min (SD = 22.32; Min. = 2; Max. = 120). A total of 52.9% of callers resulted in a follow-up appointment for another call with the same counsellor. A total of 34.9% of callers called based on a recommendation from colleagues or other referrals. Figure 1 shows the “Pathways to the PSU-HELPLINE” (“How did you hear about us?”).

### 3.3. Descriptive Data—Burden

Callers (*n* = 299) reported burden on a 5-point Likert scale (1 = not at all burdened to 5 = very burdened) with a mean of 4.00 (SD = 1.08; Min. = 1; Max. = 5). The counsellors at the PSU-HELPLINE assessed the burden of the callers (*n* = 293) on a 5-point Likert scale (1 = not at all burdened to 5 = very heavily burdened) at a mean of 3.95 (SD = 1.05; Min. = 1; Max. = 5).

The help provided by the call to the PSU-HELPLINE was rated by the participants (*n* = 281) according to their own assessment on a 5-point Likert scale (1 = not at all helpful to 5 = very helpful) with a mean of 4.04 (SD = 0.94; Min. = 2; Max. = 5). According to the counsellors’ assessment, the conversation helped the callers strongly and very strongly in 81.4% of the cases.

### 3.4. Content-Analytical Evaluation—Categories “Reasons for Calling at PSU-HELPLINE”

The reduction procedure resulted in 13 content clusters, which depict the reasons for the help-seekers’ calls, as well as 13 factors, which represent the types of impact of telephone peer counselling. The basis for the analysis of the impact factors was the five impact mechanisms of psychotherapy according to Grawe and colleagues (1994) [31].

Based on the summarising content-analytical evaluation, the following clusters could be identified as reasons for calling the PSU-HELPLINE (see Figure 2).

### 3.5. Content-Analytical Evaluation—“Effect Factors” Categories

The basis for the following categories was formed by three of the five effect factors of psychotherapy according to Grawe and colleagues (1994) [31]: “resource activation”, “problem actualisation” and “problem solving”. The aspects “therapeutic relationship” and “motivational clarification” could not be identified in the available data material. The following additional mechanisms of action could be identified (see Table 2).

## 4. Discussion

The offer of telephone counselling by peers is well accepted by the target group. The largest professional group, 30.4%, was made up of doctors. This could be explained by the results of recent studies, which show that physicians describe high levels of stress in the context of their profession, resulting in physical, psychosocial, and emotional impairment and behavioural changes [32]. Previous telephone support services have been staffed by other professions, such as psychologists, rather than by peers, and have often been focused on the target group of nurses [25,26,27,28]. The possibility to talk to a colleague seems to appeal mainly to doctors. The aspect of “real” anonymity could play a role here. This supports the findings of Bruce (2005) [33], who found that above all, trust, discretion, a protected space to communicate about problems, emotions, and doubts, etc. [33], and thus, anonymity, are important demands on support services.

The aim of this paper was to provide first results of a best practice example for psychosocial support in Germany. The following questions were posed: (1) How burdened do the callers feel and how helpful do they find the conversation at the PSU-HELPLINE? (2) What burdens do the callers mention? (3) In what way can a telephone support service reduce the psychosocial stress of the callers?

To answer the research questions, results from descriptive statistics and content analysis were used.

It was found that the burdens of the callers on a 5-point Likert scale (1 = not stressed at all to 5 = very stressed) range from MW = 4 (SD = 1.08) (assessment of the callers themselves) to MW = 3.95 (SD = 1.05) (assessment of the counsellors). The conversations are rated on a 5-point Likert scale (1 = not helpful at all to 5 = very helpful) between MW = 4.04 (SD = 0.94) (assessment of the callers themselves). The counsellors rated the calls as strongly and very strongly helpful in 81.04% of the cases. These results show, on the one hand, that the assessments of the callers and the assessment by the counsellors correspond closely in terms of content. The callers seem to call with high stress and perceive the counselling sessions as helpful. This shows the importance of providing such a support system and underlines its usefulness.

The burdens of the callers are manifold in terms of content, whereby four predominant topics could be identified. According to this, “stress due to serious events” is the most common among callers with 20.6%, followed by “stress due to the situation” (18.6%), “stress due to everyday work” (14.6%) and “private stress” (13.0%). The calls to the PSU-HELPLINE followed the presumed need in terms of content [23]. The PSU-HELPLINE was mainly used for processing serious events and in phases of overload.

The effectiveness of the PSU-HELPLINE was examined using content analysis procedures. Thirteen efficacy factors were identified (see Table 2), three of which are analogous to the efficacy factors of psychotherapy according to Grawe and colleagues (1994) [31]: “resource activation”, “problem actualisation” and “problem solving”. For the support work on the PSU-HELPLINE, further factors seem to be effective for peer counselling (on the phone) in terms of impact: change of perspective, validation, psychoeducation, containing, connectedness, information, acceptance of change, self-efficacy, emotion reduction and preservation of resources.

The following aspects are limiting: It was not possible to use data sets for analysis for all incoming calls. This is due, on the one hand, to callers who did not want to give their consent to data collection and processing, and on the other hand, to callers whose condition was assessed by the counsellors as so critical that questions on data collection could not be asked to protect the caller. Because it was not the callers themselves but the counsellors who recorded the answers of the callers, a distortion of the results (interviewer bias) cannot be ruled out. In addition, most callers (76.9%) come from southern Germany. This is mainly because the non-profit organisation PSU-Akut e.V., which offers the PSU-HELPLINE, is located in southern Germany (Munich). Measures to publicise and advertise mainly take place regionally. Regarding the reasons given for medical staff to call the PSU-HELPLINE, it should be noted that the PSU-HELPLINE was set up at the beginning of the pandemic (March 2022), which could mean that pandemic-related reasons for calling are more important than what would be the case on average over the period of use.

## 5. Conclusions

The present study contributes to getting an orienting overview of which people of the target group accept an offer of psychosocial support by telephone, which burdens the callers have and how the psychosocial support on the telephone seems to work. The results show that the needs of the target group are likely to increase in the coming years because of demographic developments and a shortage of skilled workers. Especially for employers who cannot implement an internal peer support system within the framework of staff welfare, the PSU-HELPLINE is a valuable offer. In addition to medical staff from medium to large institutions, it is also doctors and assistant professions from the smallest units, such as doctors’ practices, who can receive support through the PSU-HELPLINE, as they are unable to establish their own peer support system due to the size of the institution.

The future focus should be on making the offer better known and expanding it to be able to make this support measure accessible to as many affected healthcare workers as possible. In addition, the structures for peer support could be implemented in all healthcare facilities to be able to offer low-threshold and rapid psychosocial support internally and locally. These are not only measures in the sense of staff welfare and staff retention. Healthy medical staff are a central element for quality of care and patient safety: “Healthcare organisations should invest more time and effort in implementing evidence-based strategies to mitigate physician burnout across specialties, and particularly in emergency medicine and for physicians in training or residency” [12].

For complementary research approaches, possible specifics in the stresses due to the occupational groups could be identified in a next step. The focus of interest could be the outcome of those seeking help as well as valid evidence of effectiveness. In addition, it would be interesting to make compared analyses between gender and among follow-up callers as well as comparisons between control and intervention groups, and to conduct successive data collections over medium- and long-term periods to assess the stress suffered and its consequences on psychosocial health.

This study is part of a broader offering (e.g., also on-site support in hospitals) for medical staff to help them cope with the stresses of everyday work.

The authors are aware that this will not reduce occupational stress. Some stresses, such as serious events, are to be regarded as inherent to the profession. Nevertheless, the described support service PSU-HELPLINE can help to better cope with these stresses and to be able to provide fast and low-threshold help for one’s own colleagues when needed. In the sense of staff welfare and staff retention, further research projects could link up and examine the preservation of the ability to work and retention in the profession in long-term studies.

## Figures and Tables

**Figure 1 ijerph-19-15453-f001:**
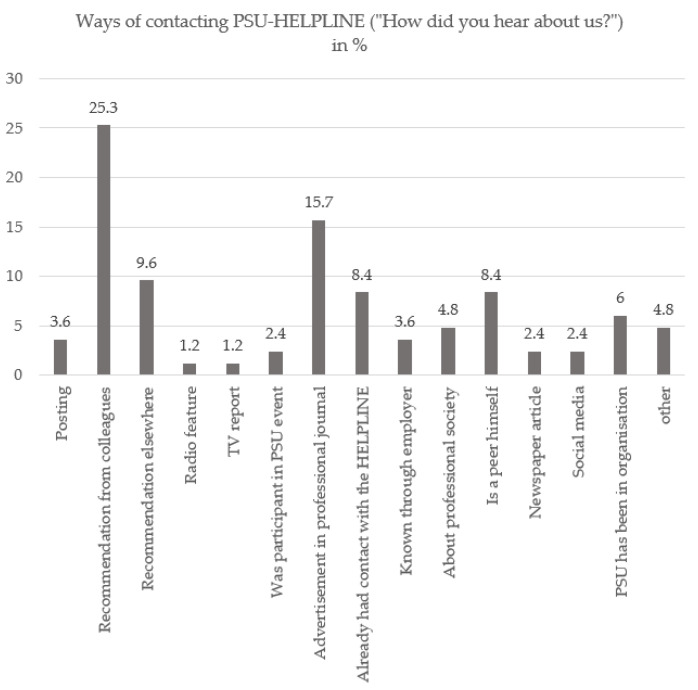
Ways of contacting PSU-HELPLINE (“How did you hear about us?”).

**Figure 2 ijerph-19-15453-f002:**
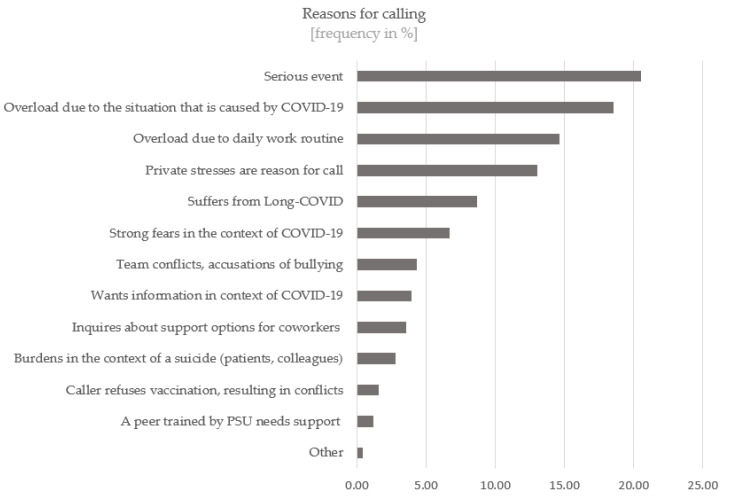
Reasons for calling at PSU-HELPLINE.

**Table 1 ijerph-19-15453-t001:** Structure of the sample.

	%	*n*
Gender (*n* = 328)		
Female	83.2	273
Male	16.8	55
Age (*n* = 217)		
<30 years	20.3	44
30–50 years	52.5	114
51–70 years	24.9	54
>70 years	2.3	5
Profession (*n* = 342)		
Nursing	24.3	83
Elderly care	18.4	63
Medical doctor	30.4	104
Medical assistant	3.8	13
Employee emergency service	1.5	5
Relative	1.2	4
Self-employed	0.9	3
Other	19.6	67

**Table 2 ijerph-19-15453-t002:** Identified effect factors and their explanation.

Effect Factor	Explanation
Psychoeducation	Understanding of own reactions is increased
Validation	Hearing and understanding that it is appropriate to feel the way you feel
Information	Knowing where to turn to for help
Preservation of resources	Things are running better again in the team as a result of the advice given to the leaders
Change of perspective	Understanding of each other is increased
Containing	Feeling accepted
Problem solving	Knowing what you can do for yourself, for the team, for the patients
Resource activation	Providing materials to the staff and developing a plan for what might do you good
Problem actualisation	Picking up and structuring issues through conversation
Acceptance of change	Reflections on not having to be perfect, not being able to have all the answers
Emotion reduction	Expressing anger, being able to vent without offending
Connectedness	To experience that the consultant understands what you are talking about
Self-efficacy	Feeling relieved because you have already thought of many things yourself, feeling confirmed and strengthened by the conversation

## Data Availability

Not applicable.

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
