# Peer review of "Calling for Help—Peer-Based Psychosocial Support for Medical Staff by Telephone—A Best Practice Example from Germany"

_ijerph, 2022, doi:10.3390/ijerph192315453_

Round 1

Reviewer 1 Report

Exciting research. Systematic method, well-argued and discussed.

Questions arise about possible gender differences in the analysis of the results and the assessment and identification of repeated calls by the same professional who asks for advice; How do you proceed in these cases? Is there a protocol of action in case of alarm due to anxiety crises or situations of very high stress? and the training in psychosocial therapy of the professionals who answer the calls? receive formal training?

It is also important to offer information about the service offered, how it is financed, and whether they are professionals, volunteers, and both.

In terms of limitations and possible future research, it would be interesting to refer to comparisons between control and intervention groups, as well as successive data collection over medium and long-term periods of time to assess the stress suffered and its consequences on psychosocial health.

Author Response

Article: Calling for Help - Peer-based psychosocial support for medical staff by telephone - a best practice example from Germany

Dear Reviewer,

Thank you very much for your review and your valuable and justified comments and suggestions.

Please find below our responses and adjustments to your comments and queries:

Comments and Suggestions for Authors

Exciting research. Systematic method, well-argued and discussed.

Comment of the authors: Thank you very much for this feedback.

Questions arise about possible gender differences in the analysis of the results and the assessment and identification of repeated calls by the same professional who asks for advice; How do you proceed in these cases? Is there a protocol of action in case of alarm due to anxiety crises or situations of very high stress? and the training in psychosocial therapy of the professionals who answer the calls? receive formal training?

Comment of the authors: Thank you for the in-depth questions. No gender differences have been examined in the analyses offered so far. We will be happy to consider and report on this aspect in further analyses. We added this aspect in chapter 5: “In addition, it would be interesting to make compared analyses between gender and among follow-up callers as well as comparisons control and intervention groups, as well as to conduct successive data collections over medium- and long-term periods to assess the stress suffered and its consequences on psychosocial health.”

The PSU-HELPLINE support provides for up to three counseling appointments to be made. If there is a further need, the callers are referred to regular psychological/ psychotherapeutic care. If necessary, the PSU-HELPLINE counselors can refer the caller in the first instance to the psychotherapeutic consultation hours on the same day (Monday - Friday). (These aspects can be found in the original paper in chapter 1.3.) In case of emergency, the PSU-HELPLINE counselors have a direct line to the psychiatric crisis service. (We added in chapter 1.3.: “In case of emergency, the PSU-HELPLINE counselors have a direct line to the psychiatric crisis service.”)

To be a counselor at the PSU-HELPLINE, many years of experience in psychosocial emergency care is a prerequisite. This can be acquired in Germany through various training institutions. Furthermore, only consultants who are themselves active in the health care sector are selected. (We added in chapter 1.3: “To be a counselor at the PSU-HELPLINE, many years of experience in psychosocial emergency care is a prerequisite. This can be acquired in Germany through various training institutions. Furthermore, only consultants who are themselves active in the health care sector are selected.”)

It is also important to offer information about the service offered, how it is financed, and whether they are professionals, volunteers, and both.

Comment of the authors: Thank you for this advice. The PSU-HELPLINE is free of charge for all callers. The service offered is financed by various health care providers in Bavaria, Germany, and by a foundation. The psychotherapists, the psychotherapeutic head as well as the coordination of the PSU-HELPLINE are employees or freelancers with approbation. The counselors of the PSU-HELPLINE work on a voluntary basis and receive an expense allowance. (We added in chapter 1.3.: “The service offered is financed by various health care providers in Bavaria, Germany, and by a foundation. The psychotherapists, the psychotherapeutic head as well as the coordination of the PSU-HELPLINE are employees or freelancers with approbation. The counselors of the PSU-HELPLINE work on a voluntary basis and receive an expense allowance.”)

In terms of limitations and possible future research, it would be interesting to refer to comparisons between control and intervention groups, as well as successive data collection over medium and long-term periods of time to assess the stress suffered and its consequences on psychosocial health.

Comment of the authors: Thank you very much for your ideas of further research. We would like to address the mentioned research areas in the future. We have included your suggestions in the conclusion, chapter 5: “In addition, it would be interesting to make compared analyses between gender and among follow-up callers as well as comparisons control and intervention groups, as well as to conduct successive data collections over medium- and long-term periods to assess the stress suffered and its consequences on psychosocial health.”

We hope you understand our adjustments, thank you again for your efforts and support, and remain available to answer any queries you may have!

Kind regards,

11/18/2022, Dominik Hinzmann, d.hinzmann@tum.de,

Reviewer 2 Report

The manuscript deals with a telephone-based support system to buffer the negative effects of the pandemic and help providers to meet the multiple needs of the population. It is well-structured and of great interest. I only suggest minor changes, as I write below:

Lines 39-40: The sentence “healthcare workers have been identified as particularly vulnerable 39 occupational groups” lacks reference.

Under the heading “Psychosocial support” (lines 57-62) there are a few sentences lacking references. Therefore, I would suggest the Author(s) improve the citations from the scientific literature.

Lines 79-80: the expressions “i.e.,” and “e.g.,” are always followed by a comma AND always go within brackets, according to the APA style 7th edition.

Line 84: according to the APA style 7th edition guidelines, “et al.,” always goes within brackets. Otherwise, it must be written “and colleagues”. See also line 87 and throughout the manuscript.

Line 195: The section “Study participants” seems to be actually poor.

However, although the “burden” of healthcare professionals is mentioned in the Introduction, it is not discussed within this project. I would suggest that the author(s) at least mention it as a relevant factor in the Discussion section.

Author Response

Article: Calling for Help - Peer-based psychosocial support for medical staff by telephone - a best practice example from Germany

Dear Reviewer,

Thank you very much for your review and your valuable and justified comments and suggestions.

Please find below our responses and adjustments to your comments and queries:

Comments and Suggestions for Authors

The manuscript deals with a telephone-based support system to buffer the negative effects of the pandemic and help providers to meet the multiple needs of the population. It is well-structured and of great interest.

Comment of the authors: Thank you very much for this feedback.

I only suggest minor changes, as I write below:

Lines 39-40: The sentence “healthcare workers have been identified as particularly vulnerable 39 occupational groups” lacks reference.

Comment of the authors: Thank you for this note. We added the following source(s) in line 40:

  1. Tyssen, R.; Vaglum, P. Mental Health Problems among Young Doctors: An Updated Review of Prospective Studies. Harv. Rev. Psychiatry 2002, 10, 154–165, doi:10.1080/10673220216218.
  2. Braun, M.; Schönfeldt-Lecuona, C.; Kessler, H.; Beck, J.; Beschoner, P.; Freudenmann, R.W. Burnout, Depression und Substanzgebrauch bei deutschen Psychiatern und Nervenärzten. Nervenheilkunde 2008, 27, 800–804, doi:10.1055/s-0038-1627220.

Under the heading “Psychosocial support” (lines 57-62) there are a few sentences lacking references. Therefore, I would suggest the Author(s) improve the citations from the scientific literature.

Comment of the authors: Thank you for this note. Here we added the following source(s) in line 65:

  1. Peterson, A.L. Experiencing stigma as a nurse with mental illness. J. Psychiatr. Ment. Health Nurs. 2017, 24, 314–321, doi:10.1111/jpm.12306.
  2. Verhaeghe, M.; Bracke, P. Associative stigma among mental health professionals: implications for professional and service user well-being. J. Health Soc. Behav. 2012, 53, 17–32, doi:10.1177/0022146512439453.

Lines 79-80: the expressions “i.e.,” and “e.g.,” are always followed by a comma AND always go within brackets, according to the APA style 7th edition.

Comment of the authors: Thank you for this note. We have transposed all passages in the text according to your comment.

Line 84: according to the APA style 7th edition guidelines, “et al.,” always goes within brackets. Otherwise, it must be written “and colleagues”. See also line 87 and throughout the manuscript.

Comment of the authors: Thank you for this note. We replaced "et al." with "and colleagues" throughout the text.

Line 195: The section “Study participants” seems to be actually poor.

Comment of the authors: Thank you for this note. We added the following in chapter 2.2: “Each call to the PSU-HELPLINE telephone number was recorded anonymously as part of their consultation. The PSU-HELPLINE counselors asked the caller whether the data could be recorded anonymously for documentation and evaluation. All persons who agreed to this procedure were included in the sample.”

However, although the “burden” of healthcare professionals is mentioned in the Introduction, it is not discussed within this project. I would suggest that the author(s) at least mention it as a relevant factor in the Discussion section.

Comment of the authors: Thank you for this comment. We added the following text in chapter 5 to refer to the “burden” again at the end: “This study is part of a broader offering (e.g., also on-site support in hospitals) for medical staff to help them cope with the stresses of everyday work. The authors are aware that this will not reduce occupational stress. Some stresses, such as serious events, are to be regarded as inherent to the profession. Nevertheless, the described support service PSU-HELPLINE can help to better cope with these stresses and to be able to provide fast and low-threshold help for one's own colleagues when needed.”

We hope you understand our adjustments, thank you again for your efforts and support, and remain available to answer any queries you may have!

Kind regards,

11/18/2022, Dominik Hinzmann, d.hinzmann@tum.de
